# MC-LoRA: Fast Modular Composition for Multi-Character Diffusion Generation

## ABSTRACT

Low-Rank Adaptation (LoRA) provides a lightweight and flexible approach for personalising diffusion models with high-fidelity characters. Yet extending LoRA to multi-character generation remains difficult: fusion-based methods require re-computing merged adapters for each character set, while non-LoRA-fusion approaches, despite avoiding image-level conditions such as pose guidance or edge maps, degrade rapidly beyond four characters, leading to scene incoherence, character vanishing, and character blending. These limitations highlight a fundamental gap: current pipelines cannot reliably scale to complex multi-character scenes while maintaining efficiency and visual quality. To address this gap, we present MC-LoRA, an inference-time framework that scales multi-character generation without retraining. MC-LoRA introduces two innovations: (i) an attention-weighted injection mechanism that balances contributions across adapters to preserve global coherence; and (ii) a dual-loss guidance scheme combining Character Balancing Loss to prevent vanishing and Spatial Localisation Loss to suppress blending. Experiments on prompts with up to eight characters show that MC-LoRA significantly outperforms LoRA-Composer, improving ImageReward from 0.046 to 0.395 during complex scenes and reducing sampling time by more than 2×. These results establish MC-LoRA as an efficient and robust solution for scalable multi-character personalisation.

## 1 INTRODUCTION

Text-to-image diffusion models have made remarkable advances in synthesising high-quality and visually compelling images from natural language prompts (Ho et al., 2020; Rombach et al., 2022). Lightweight personalisation techniques such as LoRA (Hu et al., 2021) further extend these capabilities by enabling models to learn novel concepts or characters from only a handful of examples. Since each LoRA adapter is trained independently, they can be flexibly recombined at inference, making LoRA-based pipelines both modular and scalable. While single-concept customisation with LoRA is highly effective, extending this flexibility to multi-concept generation remains a difficult challenge. When several adapters are applied simultaneously, interference between them often leads to severe artefacts, such as degraded image quality, characters vanishing (as noted in (Yang et al., 2024) and (Chefer et al., 2023)), or unintended blending between characters.

While existing works are broadly applicable to multi-concept generation, we find that for multi-character composition, the challenges of identity preservation and spatial relationships are most pronounced; furthermore, failure modes like vanishing and blending become more apparent as the number of characters increases. Existing personalisation approaches attempt to address this challenge in two main ways. Methods such as DreamBooth (Ruiz et al., 2023), HyperDreamBooth (Ruiz et al., 2024), Cones (Liu et al., 2023), and InstantBooth (Shi et al., 2023) support multi-character generation by training all concepts jointly. Yet, this dependency limits flexibility, since new concepts cannot be added without retraining. LoRA-based fusion methods (e.g., P+ (Voynov et al., 2023), Mix-of-Show (Gu et al., 2023), ZipLoRA (Shah et al., 2023), Orthogonal Adaptation (Po et al., 2024), LoRACLR (Simsar et al., 2024), OMG (Kong et al., 2024)) mitigate this restriction by merging independently trained adapters. However, these approaches require recomputing merged weights for every concept configuration, which is computationally expensive and undermines modularity. Furthermore, successful generation often depends on precise image-level conditions such as pose guidance or edge maps, limiting their accessibility and flexibility. Non-merging alternatives

| (1) Scene Incoherence | (2) Concept Vanishing | (3) Concept Blending |
|---|---|---|

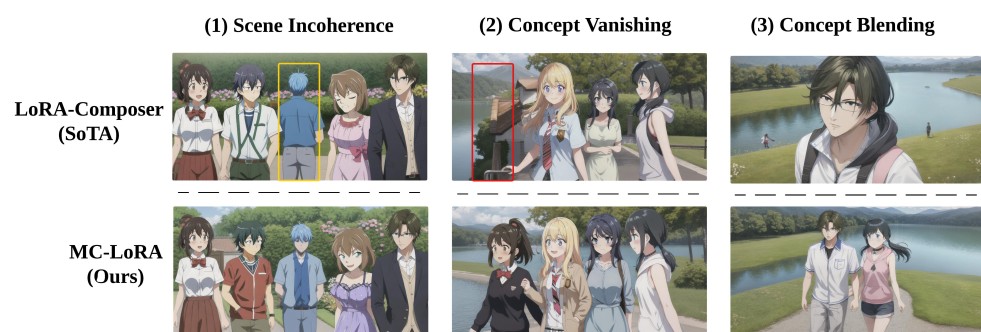

Figure 1: Illustration of the three fundamental failure modes in multi-character generation: *scene incoherence* (loss of global consistency), *character vanishing* (characters disappearing), and *character blending* (characters merging into hybrids). We compare LoRA-Composer with our proposed MC-LoRA across diverse multi-character scenes.

such as LoRA-Composer (Yang et al., 2024) omit image-level conditions and inject region-aware LoRA updates through cross-attention, avoiding repeated fusion. Nevertheless, their performance deteriorates rapidly as the number of concepts increases, leading to the disappearance or blending of characters and a loss of global scene coherence. Moreover, the inference time grows substantially with the adapter count, limiting practical deployment.

In practice, multi-character pipelines become less reliable once the number of characters exceeds four. We identify three fundamental failure modes in this regime (see Fig. 1): (1) Scene Incoherence – breakdown of global lighting or spatial consistency; (2) Character Vanishing – one or more characters fail to appear; (3) Character Blending – attributes from multiple characters merge into hybrids. Together, these issues undermine the scalability of adapter-based generation and form critical barriers to synthesising rich, multi-character scenes. These limitations stem from the architectural realities of UNet backbone diffusion models: convolutional and cross-attention layers are globally shared, allowing semantic signals from one adapter to bleed into the regions of others. Downsampling operations further exacerbate this, collapsing spatial boundaries and amplifying interference. When multiple LoRA adapters are injected simultaneously, imbalances in attention strength can cause dominant characters to overwrite weaker ones, especially in crowded scenes. Addressing these issues requires a mechanism that dynamically modulates adapter influence, enforces spatial separation, and restores balance across concepts, all without retraining.

To overcome these limitations, we propose MC-LoRA, a scalable framework that extends region-aware LoRA injection to support reliable multi-character generation. MC-LoRA introduces two complementary innovations: (i) an attention-weighted injection mechanism that dynamically rescales each adapter's contribution according to its cross-attention activations, preventing dominant characters from overwhelming weaker ones while preserving global coherence; and (ii) a dual-loss guidance scheme that operates during denoising, comprising a hinge-style Concept Balancing Loss to amplify under-attended concepts and a Tukey-window-based Spatial Localisation Loss to suppress cross-character leakage. Together, these components directly target the three common failure modes-scene incoherence, concept vanishing, and concept blending-while retaining inference-only efficiency, Assuming pre-trained character-specific adapters that are individually fused with separate copies of the base model prior to sampling, preserving modularity.

We evaluate MC-LoRA extensively across multi-character benchmarks. Our results demonstrate that MC-LoRA consistently outperforms LoRA-Composer and baseline pipelines, achieving higher fidelity, stronger coherence, and faster sampling. Notably, MC-LoRA enables more reliable generation of complex scenes containing 8 distinct characters, each occupying clearly defined regions.

Our contributions are summarised as follows:

- We propose a novel attention-weighted injection strategy that eliminates interference between adapters, preserving character fidelity and global scene coherence.
- We design a dual-loss guidance framework consisting of a Character Balancing Loss to prevent vanishing and a Spatial Localisation Loss to suppress blending.

- We present MC-LoRA, an efficient, inference-time framework for multi-character generation, and demonstrate state-of-the-art performance against existing baselines across qualitative and quantitative evaluations.

## 2 RELATED WORK

We build on recent efforts to connect multi-concept personalisation with attention-guided control in diffusion models. Methods like Attend-and-Excite (Chefer et al., 2023) address concept vanishing for generic concepts, while spatial controllers like ControlNet (Zhang et al., 2023), T2I-Adapter (Mou et al., 2023), GLIGEN (Li et al., 2023), ReCo (Yang et al., 2022), and SceneComposer (Zeng et al., 2022) achieve precise control but require additional image-based conditions such as edge maps or pose conditions.

In non-fusion LoRA composition, ComposLoRA (Zhong et al., 2024) alternates between LoRA modules without spatial control, making it prone to failure on ambiguous multi-character prompts where region-specific disambiguation is essential. Recent frequency-domain analyses such as Cached Multi-LoRA (Zou et al., 2025) reveal semantic conflicts in LoRA interactions but also operate without explicit spatial guidance. LoRA-Composer (Yang et al., 2024) integrates spatial guidance into non-fusion LoRA composition by applying region-specific updates through cross-attention, followed by self-attention masking to isolate character activations. Although effective for simpler layouts with few characters, the masking mechanism introduces substantial computational overhead. These limitations motivate our technical contributions: an attention-weighted injection strategy that moves beyond region-binary constraints, and a dual-loss guidance framework specifically designed to address the scalability challenges in multi-LoRA composition.

## 3 METHOD

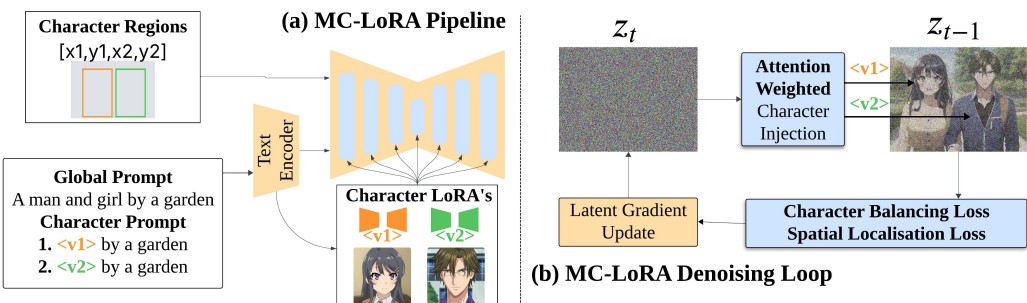

Figure 2: Overview of the MC-LoRA framework. **Left:** Multi-character pipeline showing how independently trained LoRA adapters, prompts, and bounding boxes are composed within a single diffusion process. **Right:** Per-step denoising modifications, including attention-weighted region-aware injection and dual-loss guidance (Character Balancing Loss and Spatial Localisation Loss).

Our framework operates entirely at inference time and consists of three components applied throughout the denoising process (Fig. 2): (i) an attention-weighted, region-aware LoRA injection strategy that modulates each adapter's influence based on spatial attention during the U-Net forward pass; and (ii–iii) a dual-loss guidance scheme computed after each denoising step, comprising a hinge-style character balancing loss to amplify weak activations and a Tukey-window-based spatial localisation loss to suppress cross-character leakage.

**Problem Formulation.** For each character $c_i$, we associate a LoRA adapter $\mathcal{A}_i$, a textual prompt $\mathcal{P}_i$, and a bounding box $\Omega_i = [y_1^i, x_1^i, y_2^i, x_2^i]$ specifying its spatial location. In addition, we define a background prompt $\mathcal{P}_b$ to provide global context. Collectively, for $N$ characters, we denote

$$\Omega = \{\Omega_i\}_{i=1}^N, \quad \mathcal{A} = \{\mathcal{A}_i\}_{i=1}^N, \quad \mathcal{P} = \{\mathcal{P}_i\}_{i=1}^N.$$

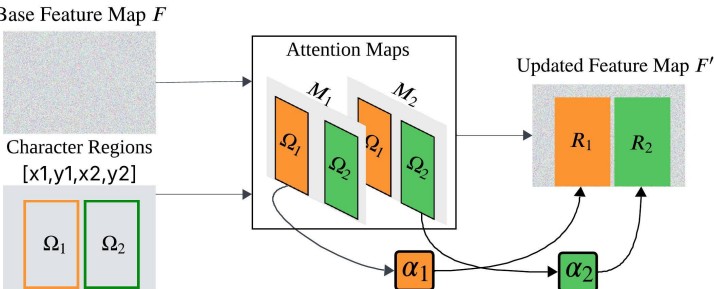

Figure 3: Attention-Weighted Region-Aware Injection. The base feature map $F$ is updated into $F'$ by adding residual updates $R_c$ from each LoRA adapter, scaled by attention-derived weights $\alpha_c$.

In the denoising process for multi-character generation, At each timestep $t$, the generated noise $\phi_t$ is computed as a function of the adapters, prompts, and spatial layout:

$$\phi_t = \mathcal{F}\big(\mathcal{A}, \mathcal{P}, \mathcal{P}_b, \Omega\big),$$

where $\phi_t$ denotes the generated noise, and $\mathcal{F}(\cdot)$ corresponds to the model conditioned jointly on global and character-level information.

**Guided Denoising Process.** Following the schedule introduced in BoxDiff (Xie et al., 2023), we refine the latent $\mathbf{z}_t$ at each timestep $t$ via a gradient descent update:

$$\mathbf{z}'_t \leftarrow \mathbf{z}_t - \phi_t \cdot \nabla \mathcal{L}, \tag{1}$$

where $\mathcal{L}$ is a constraint-driven loss function designed to regularise the denoising trajectory. The gradient $\nabla \mathcal{L}$ provides a corrective direction, guiding the denoising process to satisfy spatial and character-specific constraints. This refinement ensures both accurate localisation and strong character fidelity, while suppressing spatial interference. The updated latent $\mathbf{z}'_t$ is then passed to the U-Net for the next denoising step, ensuring that each character is synthesised within its designated region without leakage or blending.

### 3.1 ATTENTION-WEIGHTED REGION-AWARE LORA INJECTION

While spatial guidance methods such as BoxDiff (Xie et al., 2023), Attend-and-Excite (Chefer et al., 2023), and Local Conditional Controlling (Zhao et al., 2024) can help localise objects, they begin to fail in faithfully representing user-defined characters, especially in complex scenes. As the number of characters increases, attention maps become increasingly noisy, and weaker characters may receive insufficient focus, resulting in a loss of character definition, partial rendering, or complete disappearance. To address this, our method dynamically scales each adapter's contribution based on its attention presence, ensuring that all characters retain fidelity and spatial separation throughout the denoising process. This prevents dominance by any single adapter, preserves character fidelity, and promotes coherent global composition.

To quantify how strongly each character attends across all relevant cross-attention maps, we consider each prompt $P_c$ as a sequence of $L_c$ tokens in $\mathbb{R}^D$. Let $X_c(u, v, k)$ be the cross-attention weight between spatial location $(u, v)$ and token $k$ of character $c$. Averaging over tokens yields a single attention map for that character:

$$M_c = \frac{1}{L_c} \sum_{k=1}^{L_c} X_c(:, :, k), \quad M_c \in \mathbb{R}^{H \times W}. \tag{2}$$

We then measure how strongly region $\Omega_d$ is represented in each map $M_c$ by computing its mean attention:

$$\mu_{c,d} = \frac{1}{|\Omega_d|} \sum_{(u,v) \in \Omega_d} M_c(u, v). \tag{3}$$

Here, $c$ indexes the character-specific attention maps, and $d$ indexes the spatial regions associated with each character. Each character is associated with both an attention map $M_c$ and a region $\Omega_c$, and we evaluate how strongly every region $\Omega_d$ is represented in every map $M_c$. This yields a matrix of attention scores $\mu_{c,d}$, which we collect into a vector representing the attention strengths of all character maps within region $\Omega_d$:

$$\boldsymbol{\mu}_d = [\mu_{1,d}, \ldots, \mu_{N,d}] \in \mathbb{R}^N, \tag{4}$$

Let $F \in \mathbb{R}^{H_c \times W_c \times D_f}$ be the base feature map. For each concept, we compute mean attention inside its box,

$$\overline{\mu}_c = \frac{1}{|\Omega_c|} \sum_{(u,v)\in\Omega_c} M_c(u,v), \tag{5}$$

and derive injection scales via a negative softmax with temperature $T$, followed by min-normalisation:

$$\alpha_c = \frac{\exp(-\overline{\mu}_c/T)}{\min_k \exp(-\overline{\mu}_k/T) + \varepsilon}. \tag{6}$$

Each adapter $\mathcal{A}_c$ produces a residual update $R_c = \alpha_c \, \mathcal{A}_c(Q_c, K_c, V_c; F)$, which is masked to its box $\Omega_c$. The final map is

$$F' = F + \sum_{c=1}^{N} (R_c \odot \mathbb{I}_{\Omega_c}), \tag{7}$$

where $\odot$ is the element-wise Hadamard product. This formulation weights under-attended characters higher via $\alpha_c$, while confining updates to their spatial regions. Figure 3 illustrates the injection process: residuals from each adapter are selectively applied to their corresponding regions, with attention-derived weights modulating their influence. This mechanism helps preserve spatial coherence and supports balanced character representation throughout denoising.

## 3.2 CHARACTER BALANCING LOSS

When multiple adapters are injected, some characters naturally dominate due to stronger learned weights or more salient visual features. As a result, weaker characters may receive insufficient attention in their designated regions and gradually vanish over the denoising process. This phenomenon-commonly referred to as character vanishing-has been observed in prior work (Yang et al., 2024; Chefer et al., 2023), but becomes considerably more common in multi-character prompts, where increased scene complexity and overlapping attention lead to the disappearance of smaller or less distinctive entities. To counteract this, we introduce a character balancing loss that selectively boosts under-attended characters without penalising those that are already dominant.

For each concept's attention map $M_c$, we take the peak mean attention over all $N$ bounding box regions as $\beta_c = \max_{d=1,\ldots,N} \mu_{c,d}$. We then penalize the difference of the character-specific region $d = c$ from this peak through a hinge-style loss:

$$\mathcal{L}_c = \max\big(0, \; \gamma \, \beta_c - \mu_{c,c}\big), \mathcal{L}_{\text{balance}} = \sum_{c=1}^{N} \mathcal{L}_c. \tag{8}$$

Summing over all characters gives the total loss $\mathcal{L}_{\text{balance}}$. Importantly, $\beta_c$ is detached from the computational graph to prevent gradients from flowing into the dominant character, ensuring that the loss only amplifies weaker ones.

## 3.3 SPATIAL LOCALISATION LOSS

When multiple characters are generated in close proximity, attention leakage across bounding boxes can cause their spatial boundaries to blur, producing blended or overlapping features. We mitigate this by introducing a spatial localisation loss that encourages separation by constraining each character's attention to its designated region.

Each character's designated region $\Omega_d = [\, x_1^d, \, y_1^d, \, x_2^d, \, y_2^d \,]$ has width $B_d^w$ and height $B_d^h$. To softly enforce these boundaries, we construct a Tukey window profile along each axis. The Tukey window

combines a flat central region with smooth cosine-decaying edges, balancing the suppression of attention leakage and preserving character fidelity. For each axis, we define a one-dimensional Tukey window profile:

$$\mathbf{w}_d = \left[ w(0; \delta),\ w(\tfrac{1}{B_d}; \delta),\ \ldots,\ w(\tfrac{B_d-1}{B_d}; \delta) \right]^{\mathsf{T}}, \tag{9}$$

where $B_d$ is the extent along the corresponding axis. Given the attention map $M_c(u, v)$, we restrict it to $\Omega_d$ and form axis-wise predictions by taking the maximum along the orthogonal direction (e.g., $\max_x$ for vertical profiles). The tapering parameter $\delta \in [0, 1]$ controls the fraction of the profile that undergoes cosine smoothing near the boundaries-larger $\delta$ yields softer edges, while smaller values retain sharper transitions. The localisation loss $\mathcal{L}_{\text{localise}}$ is then defined as the sum of squared distances between predicted profiles and their Tukey ground-truth counterparts, over all characters and both axes.

### 3.4 Constraint-Guided Denoising Process

To ensure that each character remains spatially confined and visually distinct during denoising, we introduce gradient-based guidance applied at every timestep. The constraint loss is defined as

$$\mathcal{L} = \mathcal{L}_{\text{balance}}(\gamma) + \mathcal{L}_{\text{localise}}(\delta) \tag{10}$$

where $\mathcal{L}_{\text{balance}}$ denotes the hinge-style alignment loss (Eq. 8), $\mathcal{L}_{\text{localise}}$ is the spatial localisation loss based on Tukey window profiles (Sec. 3.3). Here, $\gamma$ and $\delta$ are hyperparameters that modulate the internal structure of the loss terms-see Eqs. (8) and (9). Importantly, this refinement operates **entirely at inference time** and requires **no additional fine-tuning** beyond pretrained LoRA adapters, thereby supporting training-free and modular composition of multiple characters.

## 4 Experiments

**Implementation** All experiments use Stable Diffusion v1.5 (Rombach et al., 2022) backbone fine-tuned on Anything-v4 (Xyn-AI) for anime-style image generation and run on a single NVIDIA H100 GPU. We evaluate our method, **MC-LoRA**, which utilises ten pre-trained LoRA adapters: five from Mix-of-Show (Gu et al., 2023), one from LoRA-Composer (Yang et al., 2024), and four custom adapters trained using the Mix-of-Show ED-LoRA framework. Each adapter is fused independently with a fresh copy of the base model prior to sampling and can be selectively loaded at run time. Hyper-parameters were optimised via a light Sobol sequence search to identify reasonable default values on a small held-out validation set prior to the main evaluation.

**Dataset** We introduce a new dataset inspired by the evaluation protocol of LoRA-Composer (Yang et al., 2024), but significantly extend it by introducing controlled variables for inter-character distance, scaling up to 8 characters, and testing robustness across distinct environments. This design results in 252 unique prompt configurations. To ensure statistical reliability, we construct a dataset by selecting three unique character combinations for each character count (ranging from 2 to 8). For each combination, we generate scenes with fixed bounding box spacing between characters (16, 32, 48, or 64 pixels). These scenes are then sampled across three distinct locations (forest, garden, lake) and evaluated under three random seeds (0, 1, 2), resulting in a total of 7×3×4×3×3=756 different scenes. This setup allows us to systematically assess performance across diverse character sets, spatial constraints, and environmental contexts.

**Evaluation** We benchmark our method against LoRA-Composer (Yang et al., 2024), the only prior work to our knowledge that supports multi-character composition via region-aware LoRA injection without requiring adapter fusion or retraining. To contextualise performance, we include visual example of a naive baseline that operates with only region-aware LoRA injection. Fusion-based approaches such as Mix-of-Show (Gu et al., 2023) and P+ (Voynov et al., 2023) are fundamentally incompatible with our setup, as they require merging adapters for each unique character configuration and depend on image-based conditions such as edge or pose maps.

We report four metrics: (1) LLM-Scoring using MiniCPM (Hu et al., 2024) inspired by (Zhong et al., 2024), which evaluates scene quality based on prompt correctness and visual aesthetics; (2)

Table 1: Evaluation results by character group. $\Delta$ denotes the absolute difference between MC-LoRA and LoRA-Composer scores (MC-LoRA − LoRA-Composer).

| Char Group | Method | MiniCPM | ImageReward | Image Alignment | Text Alignment |
|---|---|---|---|---|---|
| Low (2,3) | **MC-LoRA** | 7.1458 | **0.6782** | 0.7834 | **0.6209** |
| | LoRA-Composer | **7.1708** | 0.6249 | **0.7976** | 0.6143 |
| | $\Delta$ | -0.0250 | +0.0533 | -0.0142 | +0.0066 |
| Mid (4,5,6) | **MC-LoRA** | **6.7000** | **0.3674** | 0.7684 | **0.6117** |
| | LoRA-Composer | 6.5667 | 0.0527 | **0.7651** | 0.6093 |
| | $\Delta$ | +0.1333 | +0.3147 | +0.0033 | +0.0024 |
| High (7,8) | **MC-LoRA** | **6.6042** | **0.3952** | **0.7083** | **0.6003** |
| | LoRA-Composer | 6.3875 | 0.0466 | 0.7033 | 0.5934 |
| | $\Delta$ | +0.2167 | +0.3486 | +0.0050 | +0.0069 |

ImageReward, which estimates human visual preference calibrated to a normal distribution; we follow prior work (Gu et al., 2023) and use CLIP to measure similarity to reference images through (3) Image Alignment and (4) Text Alignment scores, representing closeness of the generated scene to prompt.

### 4.1 QUANTITATIVE RESULTS

Across all character groupings, MC-LoRA consistently outperforms LoRA-Composer on ImageReward, indicating higher perceptual quality, fewer deformities and more visually appealing generated scenes. We see similar scores for the low-character regime (2-3 characters), but MC-LoRA outperforms as the scenes get more complex, sampling visually appealing images even in the challenging mid and high-character settings, where LoRA-Composer's performance score drops sharply. Image Alignment and Text Alignment scores are close across both MC-LoRA and LoRA-Composer, suggesting that both models capture a similar degree of character-level similarity. However, MC-LoRA distinguishes itself by scoring much more strongly on ImageReward and achieving higher MiniCPM scores in the mid and high character scenes. Since MiniCPM jointly evaluates visual quality and prompt correctness, these gains indicate that MC-LoRA not only produces more coherent and aesthetically pleasing images but also adheres more faithfully to the intended prompt as scene complexity grows.

It is also important to note the limitations of CLIP-based metrics (Image Alignment and Text Alignment) in this setting. Because CLIP primarily measures global similarity, blended characters or backgrounds rendered in similar styles to the reference character can still yield artificially high alignment scores, even when individual characters are poorly generated. Likewise, CLIP text scores do not directly reflect whether each intended character is faithfully represented, but rather whether the overall scene loosely matches the textual description. For this reason, CLIP-based scores should be interpreted together with perceptual metrics such as ImageReward and MiniCPM.

### 4.2 VISUALISATION RESULTS

Figure 4 shows our generation results compared against the baseline across three different complexities in an anime style. Our method enables integration of up to eight personalised concepts, successfully alleviating the common failure modes of scene incoherence, character vanishing, and character blending. As character count increases, MC-LoRA maintains distinct identities, sharp boundaries, and coherent spatial layout, while LoRA-Composer exhibits degraded fidelity and frequent blending. The naive baseline, which uses vanilla Stable Diffusion with only region-aware injection, fails to preserve character identities and lacks scene consistency. All examples are generated using identical prompts, seeds, and inference settings, highlighting the robustness of MC-LoRA in complex multi-character scenes.

### 4.3 SAMPLING TIME

Below we report the inference latency corresponding solely to the sampling phase and excluding model loading overhead. Which took approximately 3 s per LoRA adapter plus 3 s for the base

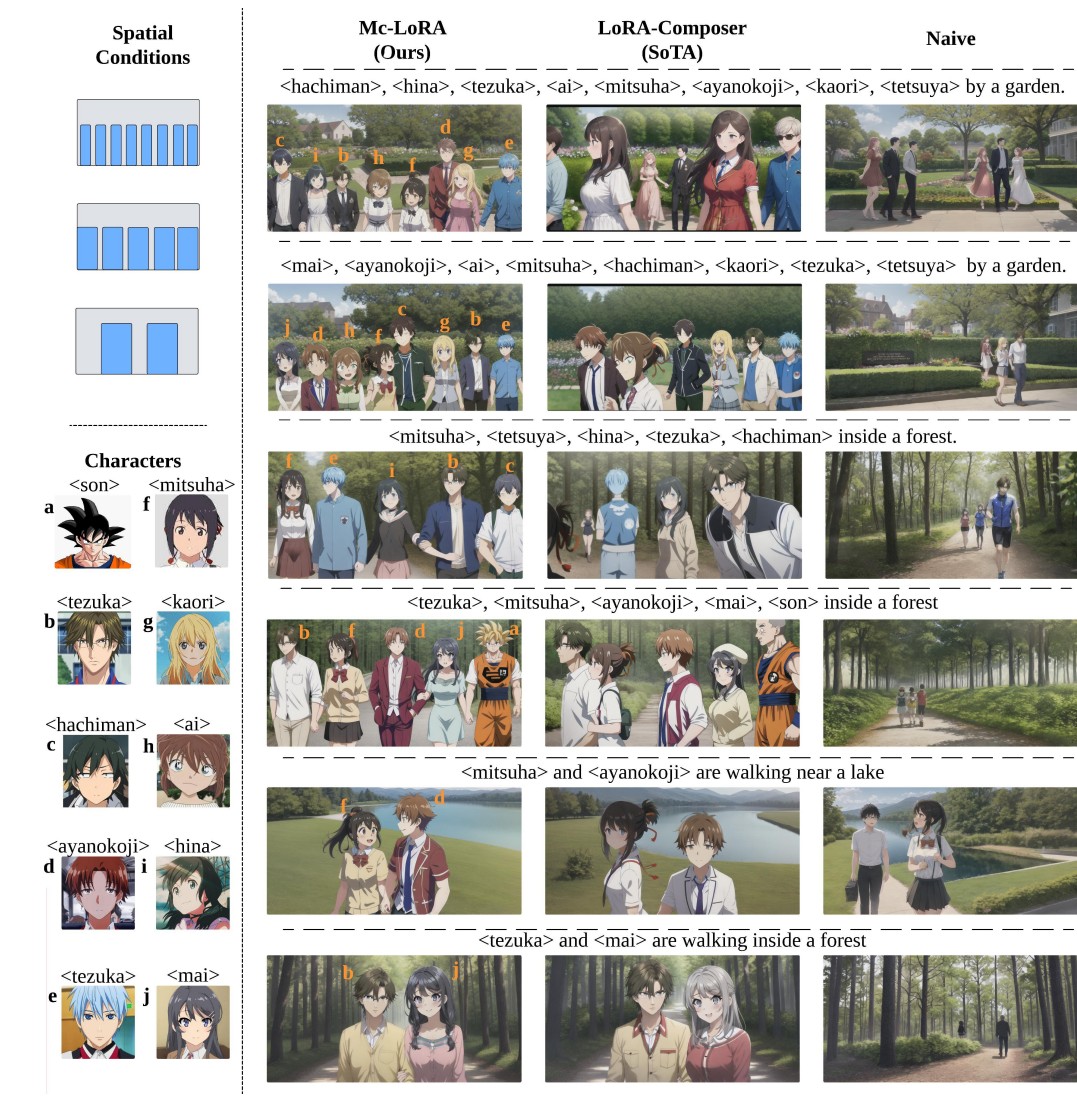

Figure 4: Qualitative comparison of multi-character generation with 2, 5, 8 characters. Intended characters are labelled in MC-LoRA column.

Table 2: Average inference time (s) per **character number**, excluding model loading time. Values in parentheses indicate relative speedup compared to the Naive baseline (*higher is better*).

| Method | 2 | 3 | 4 | 5 | 6 | 7 | 8 |
|---|---|---|---|---|---|---|---|
| Naive | 7.40 | 8.04 | 8.66 | 9.32 | 9.94 | **10.63** | **11.19** |
| LoRA-Composer | 13.45 (0.55×) | 15.05 (0.53×) | 18.15 (0.48×) | 19.97 (0.47×) | 22.68 (0.44×) | 24.37 (0.44×) | 26.88 (0.42×) |
| **MC-LoRA** | **6.74 (1.10×)** | **7.71 (1.04×)** | **8.38 (1.03×)** | **9.24 (1.01×)** | **10.39 (0.96×)** | 10.74 (0.99×) | 11.42 (0.98×) |

model. We also include the naive baseline, and show that our added contributions can have a small effect of increasing sampling speeds for Low and Middle character groups (two to six character scenes) and have negligible added overhead even when scenes grow complex, comparing against a vanilla Stable Diffusion model with only Region-Aware Injection.

As shown in Table 2, MC-LoRA delivers a substantial reduction in sampling time for every character count, with inference times less than half of LoRA-Composer's. Crucially, MC-LoRA's runtime scales slowly with the number of characters, increasing by only ∼4.7 seconds from 2 to 8 characters, whereas LoRA-Composer's runtime more than doubles over the same range. This confirms that our

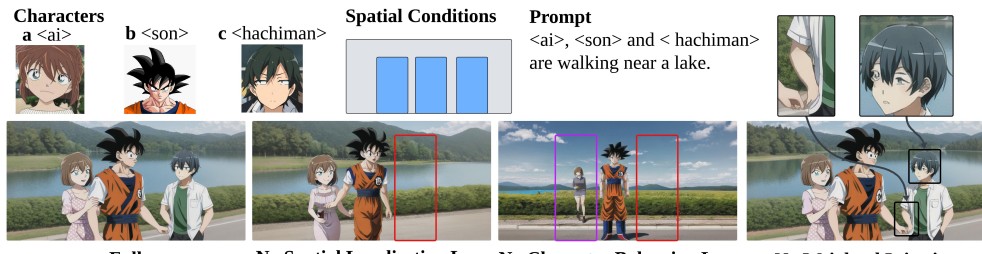

Figure 5: Ablation study of MC-LoRA, showing results with and without each component, all across the same seed and input conditions.

Table 3: Evaluation results by character group

| Method | MiniCPM | ImageReward | Image Alignment | Text Alignment |
|---|---|---|---|---|
| **MC-LoRA** | 6.57 | 0.461 | 0.755 | 0.613 |
| Without Weighted Injection | 6.20 | 0.495 | 0.758 | 0.613 |
| Without Character Balance Loss | 5.99 | 0.282 | 0.744 | 0.622 |
| Without Spatial Localisation Loss | 6.27 | 0.241 | 0.700 | 0.632 |

novel method including Attention-Weighted Region Aware Injection, the dual-loss framework and the removal of self-attention masking enables scalable multi-character generation with negligible computation overhead.

## 4.4 ABLATION

We conduct an ablation study on a reduced dataset one-third the size of our main dataset, reporting both quantitative metrics (Table 3) and qualitative comparisons (Fig. 5). Without Spatial Localisation Loss character $c$ disappears entirely. Although designed to prevent character blending, this loss also plays an additional role in maintaining character presence and avoiding vanishing artefacts, as seen by a drop in ImageReward and Image Alignment scores. Without Character Balancing Loss character $c$ vanishes, while character $a$ is pushed into the background and no longer resembles the reference. This degradation likely stems from character $b$ dominating the attention landscape, suppressing weaker characters during denoising. These failures highlight the importance of balanced attention to ensure all characters remain visible, distinct, and faithful to the prompt. Without Attention-Weighted Region-Aware Injection overall composition is preserved, but fine-grained artefacts emerge, such as distortions in character $c$'s eyes and arm. These issues are subtle and do not substantially impact alignment metrics or ImageReward scores, which primarily reflect global coherence and prompt fidelity rather than local detail. This highlights the importance of region-aware injection for preserving fine-grained character features and ensuring coherent multi-character scenes.

## 5 CONCLUSION

We introduced **MC-LoRA**, a modular and scalable framework for multi-character generation that achieves high-fidelity results without relying on image-based guidance. Our method supports up to eight personalised characters within a single scene, preserving character layouts, identities, and fine-grained details across diverse compositions. Through extensive experiments, we demonstrate that MC-LoRA consistently outperforms prior baselines in visual quality, prompt adherence, and scene coherence. Moreover, it achieves faster sampling than the naive baseline for low to mid character counts (two to six), with only negligible overhead when scaling to seven or eight characters. These results highlight MC-LoRA's practicality for complex multi-character generation in real-world applications.

ETHICS STATEMENT

The pretrained LoRAs employed in this research may reflect biases or generate sensitive or potentially offensive content, intended solely for academic and scientific purposes. The opinions expressed within generated outputs do not represent the views of the authors. We remain committed to fostering the development of AI technologies which align with ethical standards and reflect societal values.

REPRODUCIBILITY STATEMENT

We detail our work in the Method section (Sec. 3) and describe implementation details in the Experiments section (Sec. 4).

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

## A  STATEMENT ON LLM USAGE

We acknowledge that large language models (LLMs) were used to assist with polishing the writing of this paper. All research ideas, methods, and experimental results presented are original contributions of the authors.

