# OpenReview forum: "MC-LoRA: Fast Modular Composition for Multi-Character Diffusion Generation"
_ICLR.cc/2026/Conference — ICLR 2026 Conference Withdrawn Submission_

### Official Review · Reviewer_c9WV · 2025-10-16

**Soundness:** 3
**Presentation:** 3
**Contribution:** 3
**Rating:** 4
**Confidence:** 5

**Summary:**

This paper proposes MC-LoRA, a training-free, inference-time framework designed to enable efficient and scalable multi-character generation using LoRA (Low-Rank Adaptation) adapters in diffusion models. MC-LoRA introduces two main contributions: 1. Dynamically adjusts the influence of each character’s LoRA adapter based on cross-attention activations, ensuring balanced representation and preventing dominance or interference among characters. 2. A combination of Character Balancing Loss and Spatial Localisation Loss. These mechanisms act during denoising to preserve spatial boundaries and character fidelity. Experiments on an extended multi-character dataset demonstrate that MC-LoRA significantly improves visual quality, scene coherence, and sampling speed compared with LoRA-Composer, the previous state-of-the-art non-fusion approach.

**Strengths:**

1. The entire method operates at inference time, preserving LoRA modularity, allowing flexible composition of independently trained adapters without retraining or merging.
2. Each component’s contribution is isolated, and failure cases are clearly visualized, demonstrating solid experimental rigor and interpretability.
3. The framework reduces inference latency by more than 2× compared with LoRA-Composer and scales gracefully with character count, maintaining practical runtime even for complex 8-character scenes.

**Weaknesses:**

1. The approach assumes pre-defined bounding boxes for each character, which simplifies the spatial separation problem. Real-world prompts without explicit layout control may degrade performance.
2. Only LoRA-Composer is used as a baseline. While justified for the non-fusion setting, it would strengthen the paper to include comparisons to recent compositional diffusion methods even if indirectly.
3. Although the paper demonstrates success up to 8 characters, it does not analyze the limitations beyond this range or under extreme layout overlaps.
4. The attention-weighted scaling and loss functions are heuristically motivated. The paper could benefit from theoretical justification or a deeper discussion of convergence behavior.

**Questions:**

See weakness

---

### Official Review · Reviewer_YgPU · 2025-10-30

**Soundness:** 3
**Presentation:** 3
**Contribution:** 3
**Rating:** 4
**Confidence:** 3

**Summary:**

This paper introduces MC-LoRA, an inference-time framework designed to efficiently and coherently generate multi-character scenes using independently trained LoRA adapters. The key challenge addressed is that existing LoRA fusion or non-fusion pipelines struggle to scale beyond four characters due to scene incoherence, character vanishing, and character blending. To overcome this, MC-LoRA proposes two technical contributions Attention-Weighted Region-Aware Injection and Dual-Loss Guidance Framework. Extensive experiments on anime-style diffusion models (Stable Diffusion v1.5 + Anything-v4) demonstrate that MC-LoRA achieves superior ImageReward and MiniCPM scores compared to LoRA-Composer, while being 2× faster in sampling. The framework supports up to 8 distinct characters within a single coherent scene, outperforming prior modular and fusion-based approaches.

**Strengths:**

- The method bridges gaps between region-aware LoRA composition and attention modulation, which seems to be a combination not explored in prior literature.
- The paper is well-organized and accessible.

**Weaknesses:**

- The evaluation could be more rigorous; it currently compares the proposed method only to a baseline, without benchmarking against other competitive counterparts.
- The ablation results could more clearly highlight the benefit of the proposed module. For example, in the “Without Weighted Injection” case, performance on most metrics is comparable to or even higher than that of the complete version, suggesting that further justification may be needed.

**Questions:**

- The framework also seems to convert the bounding box to a mask representation. What's the difference between this and Lora-Composer?
- Could you please provide more visualization results. In addition, a complete user study to verify the performance of the proposed method is particularly significant, as the benchmarking strategy is relatively limited and incomplete.

---

### Official Review · Reviewer_MVjZ · 2025-10-30

**Soundness:** 2
**Presentation:** 2
**Contribution:** 2
**Rating:** 2
**Confidence:** 5

**Summary:**

MC-LoRA addresses the challenge of fidelity degradation in personalized diffusion models when generating multiple subjects (characters), a problem that becomes more severe as the number of subjects increases. The authors propose a training-free framework that reportedly achieves scalable multi-subject generation by introducing two key mechanisms: (1) an attention-weighted injection mechanism that modulates the cross-attention blocks within LoRA weights to manage coherence between the character-specific adapter and the base model's global information, and (2) a dual-loss guidance (character balancing and spatial localization) applied as an inference-time optimization to correct the latent representation. The authors claim this method robustly generates up to eight distinct characters without degradation.

**Strengths:**

- Direct Latent Control: The introduction of explicit, human-interpretable loss functions (e.g., spatial localization) for inference-time guidance is a straightforward approach to enforcing user intent, particularly for placing subjects within specified bounding boxes.

- Spatial Blending Technique: The use of a Tukey window to apply spatial guidance is a sensible technique, as it helps mitigate the hard-edge artifacts common in box-based constraints and promotes more natural blending with the background.

**Weaknesses:**

This paper, in its current form, suffers from fundamental flaws in its methodology, evaluation, and analysis. The core contribution is undermined by a critical misunderstanding or omission of the trade-offs involved, and the experimental validation is insufficient to support the claims.
1.  Misalignment with LoRA's Core Benefits (Sacrificing Efficiency): My primary concern is that the proposed method is fundamentally misaligned with the "LoRA" framework it claims to use. A core, defining advantage of LoRA is its ability to be statically merged into the base model, resulting in zero inference overhead. The proposed method, which relies on complex, runtime interventions (attention injection and iterative latent optimization via dual losses), completely forfeits this advantage. This is a massive trade-off that is neither acknowledged nor discussed.

2. Inference Overhead : This critical flaw is made explicit in the paper's own "Sampling Time" results.
 - First, the table (Table 2) appears to contain a significant typo, claiming "higher is better" for sampling time, which is nonsensical.
 - Second, and more importantly, the results clearly show that sampling time scales linearly with the number of characters. This is the strongest possible evidence that the method is not a "LoRA" solution in the practical sense, but rather a computationally expensive, adapter-based guidance technique. The authors position the work as a "LoRA" method but ignore LoRA's primary benefit.

3. Severe Lack of Analysis and Ablation: For a paper that claims to "manage coherence" in cross-attention, it provides zero analysis of its core mechanism.
  - No Mechanism Analysis: There are no visualizations, attention maps, or intrinsic analyses to show how the attention-weighted injection actually works, what effect it has on the cross-attention layers, or how it prevents character leakage.
  - No Ablation Study: The paper lacks even the most basic ablations. For instance:
    1). Does the guidance need to be applied at all timesteps? A timestep-dependent analysis is missing.
    2). What is the relative impact of intervening in conv blocks versus cross-attn blocks?
    3). How do the hyperparameters for the dual-loss (e.g., loss weights) affect the balance between fidelity and localization?

4. Insufficient and Poorly-Justified Evaluation: The experimental setup is weak and raises more questions than it answers.
  - Domain Bias: All experiments are conducted only on the "Anything V4" (anime-style) model. There is zero evidence that this approach generalizes to other base models (e.S., photorealistic models like SDXL) or other scenarios.
  - Statistical Significance: Results are presented without standard deviations or error bars. It is impossible to know if the reported gains are statistically significant or merely due to random seed variations.
  - Problematic VLM Judge (MiniCPM): The use of MiniCPM as a "judge" is highly suspect. Evaluating spatial localization and fine-grained attributes is a challenging task even for state-of-the-art VLMs. The authors provide no justification that MiniCPM is a reliable or accurate evaluator for this specific task. Furthermore, the prompts, templates, and hyperparameters used to query this VLM are not provided, making the evaluation irreproducible.

**Questions:**

The weaknesses listed above are severe. A convincing rebuttal must address the following points directly:

### Questions
1. Generalizability: Can you provide any evidence that this method generalizes beyond the single "Anything V4" anime model?

2. Significance: Can you provide results with standard deviations or error bars to demonstrate that the improvements are statistically significant?

3. CLIP Metric: Please clarify precisely how the CLIP metric is being used to evaluate fidelity to unseen personalized concepts. What is the text prompt, and what is the image it is compared against?

4. VLM Judge: Please provide justification (e.g., citations, preliminary validation experiments) for using MiniCPM as a reliable judge for spatial localization. Please also provide the exact prompts, templates, and hyperparameters used for this VLM evaluation.

5. Analysis: To support your claims, can you provide any analysis of the internal mechanisms, such as attention maps (to show character localization/leakage) or a timestep ablation study?

6. Typo: Please confirm the typo in the "Sampling Time" table (i.e., it should be "lower is better").


### LLM Disclosure
I have used an LLM to assist with improving the grammar, clarity, and polishing of this review. The content, analysis, and final judgments are entirely my own.

---

### Official Review · Reviewer_ZXAQ · 2025-10-31

**Soundness:** 2
**Presentation:** 1
**Contribution:** 2
**Rating:** 2
**Confidence:** 3

**Summary:**

The paper proposes MC-LoRA, an inference-time framework designed to improve multi-character image generation using diffusion models with multiple LoRA adapters. The method introduces an attention-weighted injection mechanism to balance the influence of different adapters and a dual-loss guidance scheme—comprising a Character Balancing Loss and a Spatial Localisation Loss—to mitigate issues such as character vanishing, blending, and scene incoherence. MC-LoRA operates without retraining and aims to maintain modularity and efficiency during inference. Experimental results report improved visual coherence and faster sampling compared to LoRA-Composer, with evaluations conducted on synthetic multi-character benchmarks.

**Strengths:**

The paper addresses an interesting and practically relevant problem—scaling LoRA-based diffusion models to multi-character generation without retraining—which is both novel and challenging. The proposed inference-time composition framework is original in combining attention-weighted adapter injection with a dual-loss guidance mechanism, representing a creative attempt to improve spatial and identity consistency across multiple characters. The formulation is clearly presented, with structured sections and detailed descriptions of the method and experimental setup. While primarily empirical, the work contributes a new perspective on modular composition and inference-time control in diffusion models, and may hold significance for future research on scalable personalization and compositional generation.

**Weaknesses:**

While the paper presents an interesting idea, its main weaknesses lie in the lack of theoretical grounding and limited experimental validation. The proposed inference-time loss guidance is largely heuristic, and the paper does not clearly justify how the introduced gradients interact with the diffusion denoising process or ensure stability. The evaluation omits stronger baselines that involve multi-concept training, such as Cones or Mix-of-Show, making it difficult to assess relative performance. Reported improvements rely mainly on subjective metrics, while CLIP-based scores are acknowledged to be insensitive to character-level errors. In addition, the framework requires manual bounding boxes and prompt design, which limits its practical modularity. A deeper theoretical analysis, more rigorous quantitative evaluation, and inclusion of broader baselines would strengthen the contribution.

**Questions:**

1. Clarification is needed on how the proposed loss gradients are computed and propagated through the denoising process, including their effect on sampling stability and convergence.

2. The theoretical basis of the inference-time loss guidance remains unclear, and a formal analysis connecting the proposed constraints to the underlying diffusion objective would strengthen the claim of principled design.

3. The experimental comparison should include control-based baselines such as ControlNet, which also handle spatial conditioning and multi-entity composition, to better contextualize the proposed method’s advantages.

4. The evaluation would benefit from including other baseliens (e.g., K-LoRA) to more comprehensively assess scalability and visual quality.

5. The reliance on manually defined bounding boxes and character prompts limits the method’s automation and modularity, and further clarification on how these could be inferred or learned automatically would improve practicality.

---

### Note · Authors · 2026-01-18

I have read and agree with the venue's withdrawal policy on behalf of myself and my co-authors.